



# Energetics of monsoons and deserts: role of surface albedo vs water vapor feedback

Chetankumar Jalihal[1,2] and Uwe Mikolajewicz[1]

[1]Max Planck Institute for Meteorology, Hamburg
[2]Indian Institute of Technology Hyderabad

**Correspondence:** Chetankumar Jalihal (chetankumar.jalihal@mpimet.mpg.de)

**Abstract.** Despite receiving similar solar energy, the top-of-the-atmosphere (TOA) radiation budget is negative over deserts and positive over monsoon domains. This contrast has been attributed to differences in the surface albedo between the two regions. Here, we show that this difference in TOA radiation budget is primarily driven by the absorption of longwave radiation by water vapor, while the surface albedo only plays a secondary role. As a greenhouse gas, water vapor absorbs the surface longwave radiation and enhances the local radiative heating of the atmosphere. Due to the aridity of the deserts and prevalent clear sky conditions, long wave energy is efficiently radiated to space. We demonstrate that this is the dominant cause of the net radiative cooling of the atmosphere. Our hypothesis is confirmed by a climate model experiment in which the Earth's rotation is reversed. This flips the zonal asymmetries producing a monsoon over the Sahara (in spite of high albedo) and a desert over South East Asia (where low albedo persists). We find that radiative feedback of water vapor on the large-scale circulation drives these changes initially, with further amplification by cloud feedbacks. Our results suggest that this radiation-circulation feedback due to water vapor enhances monsoon-desert contrast.

## 1 Introduction

During boreal summer, the top-of-the-atmosphere (TOA) radiation budget over the Sahara is negative, which is in stark contrast to the positive TOA radiation budget observed over the South Asian monsoon region (Charney, 1975; Wallace and Hobbs, 2006; Dewitte and Clerbaux, 2017). This contrast has been largely attributed to the high surface albedo of the Sahara, where bright sand reflects a significant portion of incoming solar radiation (Charney, 1975). However, the TOA radiation budget is also influenced by factors such as atmospheric water vapor content, cloud cover, and cloud-top height. Clouds emit longwave radiation at cooler temperatures than the Earth's surface. The height of cloud tops and the extent of cloud cover significantly affect the outgoing longwave radiation (OLR). In cloud-free regions, surface temperature and the amount of water vapor play an important role. Water vapor, a potent greenhouse gas, absorbs longwave radiation emitted by the surface and re-emits it from higher, cooler altitudes. Deserts are characterized by clear skies, dry air, and high surface temperatures. Thus, deserts tend to emit higher levels of OLR than monsoons (Wallace and Hobbs, 2006) (Fig. 1). While the influence of these factors is recognized (Charney, 1975; Rodwell and Hoskins, 1996; Alamirew et al., 2018; Wallace and Hobbs, 2006), a quantification of their relative importance on the monsoon-desert TOA radiation budget contrast is missing and the major focus in the literature



has been on the differences in surface albedo. Besides, the zonal deviation of the TOA radiation budget is primarily influenced by the OLR rather than the reflected shortwave radiation, both in monsoon and desert regions (Fig 1). A mechanism for this has not yet been addressed. During the non-monsoon months, the TOA radiation budget is negative over both the Sahara and the South Asian monsoon region (Supplementary Fig. 1a). In the monsoon months of June, July, and August, OLR plays the largest role in the TOA radiation budget contrast between the Sahara and the South Asian monsoon region (Supplementary Fig.

1b & 1c), leading to a positive radiation budget over the South Asian monsoon region. Thus, further underlining the role of OLR on the radiation budget contrast between monsoons and deserts. However, this has not yet been investigated in literature.

     The sequence of events that creates these differences between monsoons and deserts remains largely speculative and lacks substantial evidence. The diabatic heating associated with monsoons triggers adiabatic subsidence to the west of the monsoon regions (Rodwell and Hoskins, 1996) (Fig. 1). It is suggested that this reduced relative humidity initially inhibits con-

vection, promotes clear skies, and increases OLR. Over time, the resulting arid conditions reduce vegetation cover, which in turn increases surface albedo (Knorr and Schnitzler, 2006; Zeng and Yoon, 2009). The higher albedo enhances radiative cooling, further strengthening subsidence (Zeng and Neelin, 2000; Bonan, 2008). This vegetation-albedo feedback, initiated by monsoon-driven subsidence, is thought to be the key process leading to the differences in the TOA radiation budget between monsoon and deserts regions (Rodwell and Hoskins, 1996, 2001; Chou and Neelin, 2003). However, recent studies

have demonstrated that deserts can also impact monsoons (Vinoj et al., 2014; Chakraborty and Agrawal, 2017; Sooraj et al., 2019, 2021; Pausata et al., 2021; Singh and Sandeep, 2022). For example, vegetation changes and dust emissions from deserts can modulate the South Asian monsoon (Vinoj et al., 2014; Pausata et al., 2021). Additionally, dry air intrusion from deserts impacts monsoon dynamics (Parker et al., 2016; Singh and Sandeep, 2022; Rai and Raveh-Rubin, 2023). Thus, monsoons and deserts engage in a two-way interaction that manifests as differences in the TOA radiation budget in a steady state. A delin-

eation of the pathway through which parameters that influence TOA radiation budget contrast between monsoons and deserts in comprehensive Earth system models has not been realized so far. In this study, we examine the steady state as well as the evolution of TOA radiation budget contrast between monsoons and deserts. We chose a simulation where the rotation of the Earth is reversed (RETRO) (Mikolajewicz et al., 2018). In this simulation the Sahara becomes monsoonal and subsidence (and thus an arid climate) develops over the South East and East Asia. This simultaneous reorganization of monsoons and deserts

enables a thorough examination of the pathway through which the TOA radiation budget contrast emerges over monsoons and deserts.

     A common framework for monsoons and deserts is required for this analysis. However, different theories exist for monsoons and deserts. The TOA radiation budget over land is balanced by the horizontal and vertical advection of moist static energy (MSE) (Trenberth and Smith, 2009). The monsoon circulation balances the positive TOA radiation budget by exporting MSE

from the monsoon domain (Neelin and Held, 1987; Biasutti et al., 2018; Hill, 2019). The advection of MSE by monsoons is related to the strength of convection through gross moist stability (GMS) (Srinivasan, 2001; Raymond et al., 2009; Jalihal et al., 2019a). GMS is, therefore, the efficiency of monsoon convection in exporting MSE. This framework is called the energetics of monsoons (Biasutti et al., 2018; Hill, 2019). The energetics framework can be extended to the deserts (Neelin and Held, 1987). Deserts import MSE (for deserts this is nearly equal to dry static energy) which balances the negative TOA radiation budget.





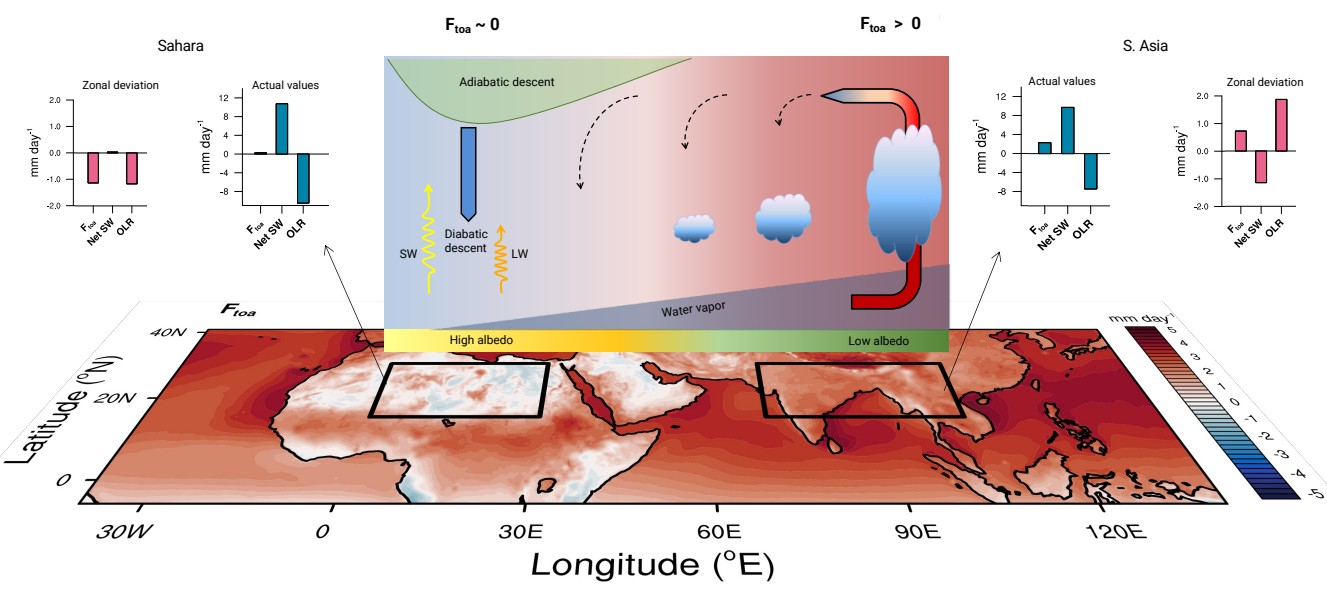

**Figure 1. Schematic of monsoon-desert mechanism.** The spatial plot depicts the Jun-Jul-Aug (JJA) climatology of TOA net radiation budget ($F_{toa}$) from ERA-5 (1981-2020). The bar charts in blue show the $F_{toa}$ and its components, the net TOA shortwave (Net_SW) and outgoing longwave radiation (OLR) (see Appendix A4) area averaged over the regions enclosed by the thick black box. The bar charts in pink represent the same quantities, albeit after removing the zonal mean from land-only grids. The schematic shows a vertical cross-section of the atmosphere.

The horizontal advection of MSE must be taken into account (Tyrlis et al., 2013; Cherchi et al., 2014; Jalihal et al., 2019a). GMS over the deserts can be interpreted as the efficiency of the subsidence in importing MSE. As energetics accounts for the TOA radiation budget and circulation, it serves as a common framework for understanding monsoons and deserts. In this study, we have used the ERA-5 reanalysis because it includes all the variables required for calculating GMS and the radiation budget. The $F_{toa}$ from ERA-5 closely matches CERES data (Supplementary Fig. 2).

## 2  Results


The profiles of vertical velocity, specific humidity, and MSE are crucial in determining the value of GMS (Srinivasan, 2001; Inoue et al., 2021). The profiles of vertical velocity and MSE differ significantly between the Sahara and the South Asian monsoon (Supplementary Fig. 3a & 3c). However, our diagnostics suggest that the differences in GMS between the South Asian monsoon and the Sahara do not explain the contrast in energetics between the two regions and nearly all of the contrast



in the energetics is due to $F_{toa}$ (Fig. 2a). This is mainly due to OLR (Fig. 2b). The contrast in reflectivity has a secondary impact on $F_{toa}$. Further decomposing OLR (see Appendix A4) suggests that the clear sky OLR and cloud cover are the prominent terms. Shown in Fig. 2c and 2d is the relation between the clear sky OLR and surface temperature, and clear sky OLR and column water vapor (CWV), respectively. Over the Sahara, clear sky OLR is essentially a linear function of surface temperature. Higher temperatures lead to larger clear sky longwave emissions. CWV over the Sahara does not exceed 25 kg m$^{-2}$, and does

not influence the clear sky OLR. CWV has a large range in the South Asian monsoon domain. During the non-monsoon months, CWV is low and clear sky OLR is a linear function of surface temperature, similar to the Sahara. As CWV increases during the onset and progression of the monsoon, clear sky OLR transitions from a state where it depends on surface temperature to a state where it depends on CWV. Thus, radiative effect of moisture is the leading cause of difference in clear sky OLR, and $F_{toa}$, between South Asia and the Sahara.

80       To understand the energetics of monsoons and deserts under a different circulation scenario, we use CTL (the pre-industrial control simulation) and RETRO simulations from a fully coupled Earth system model (see Appendix A1). In RETRO, Rossby waves propagate to the east. Thus, in RETRO a large desert exists over the South East and East Asia, to the east of monsoon which is now prevalent over the Sahara (Fig. 3a). This redistribution of monsoons and deserts aligns with the predictions of the Rodwell-Hoskins mechanism (Rodwell and Hoskins, 1996, 2001). $F_{toa}$ in RETRO is positive over the monsoonal Sahara

and also over South East and East Asia which is a desert in RETRO (Fig. 3c). This carries two implications, one for monsoons and one for deserts. Positive $F_{toa}$ is a necessary but not a sufficient condition for monsoons to exist. Deserts on the other hand persist under both positive and negative $F_{toa}$ conditions. This suggests that dynamics and not radiation is the driving factor for desertification.

       The change in moisture convergence over the RETRO Sahara is almost entirely driven by changes in $F_{toa}$ (Supplementary

Fig. 4a). The OLR is the primary cause of the increase in $F_{toa}$ over the RETRO Sahara (Supplementary Fig. 4b). The reflected shortwave has remained nearly constant. This is because the surface albedo of the desert Sahara is nearly as high as that of the cloud-covered Sahara. The changes in OLR are related to clear sky radiative effects of water vapor (Supplementary Fig. 4b, 4c, & 4d). Substantial moisture advection into the Sahara from the east (South Asian monsoon winds) and the west (anticyclone associated with the Atlantic subtropical high) leads to higher CWV in RETRO (Fig. 4a & 4b). Thus, the clear

sky OLR over the Sahara is driven by different factors in CTL and RETRO. The demise of monsoon in South East and East Asia in RETRO is largely due to the changes in GMS (Supplementary Fig. 5), emphasizing the role of atmospheric dynamics. The strengthening of the monsoon over the RETRO Sahara during the boreal summer induces a subsidence over the South East and East Asia (Fig. 4c). This adiabatic subsidence is related to the Rossby wave triggered by the diabatic heating over the RETRO Sahara (Supplementary Fig. 6a & 6b). With the withdrawal of the monsoon, the adiabatic subsidence over South East

and East Asia weakens. The mid-latitude storms penetrate into this region producing precipitation during the boreal winter and vegetation proliferates reducing albedo. In CTL, the subsidence over the Sahara increases with the onset of the South Asian monsoon. However, the subsidence continues to persist during non-monsoonal months. This subsidence during the non-monsoonal months is due to the diabatic descent driven by radiative cooling. Our result suggests that the albedo of desert perhaps is important in maintaining subsidence during the non-monsoonal months when the adiabatic descent is absent.





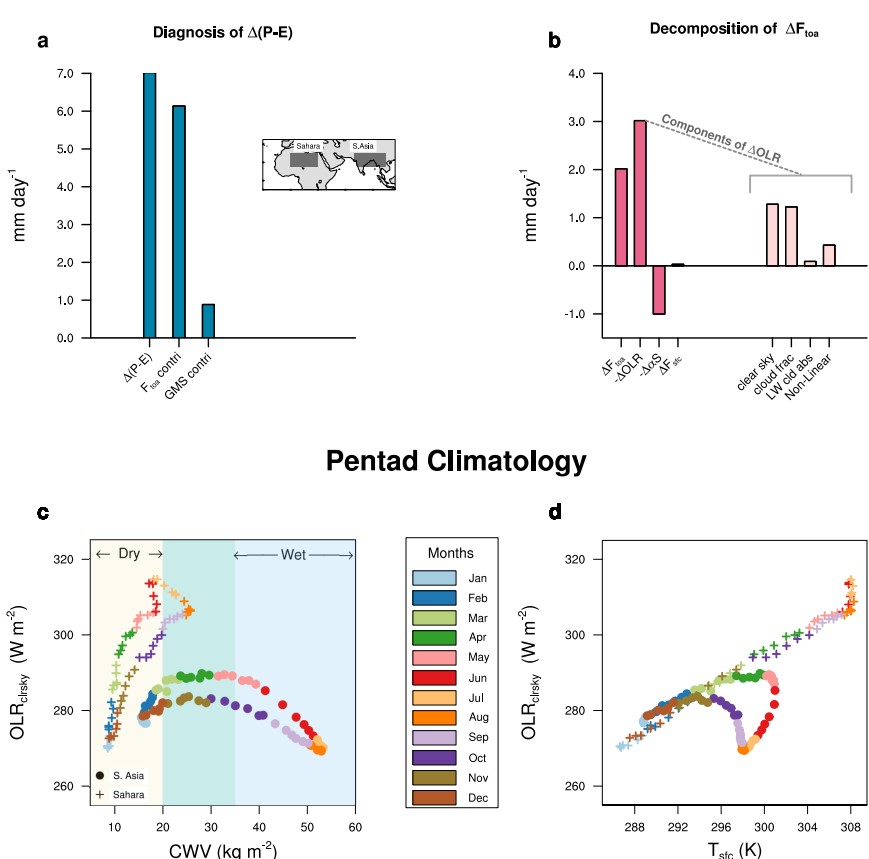

**Figure 2. Diagnosis of difference in moisture convergence between Sahara and South Asia** Bar graph of (**a**) the difference in Precipitation minus Evaporation (P-E) between the Sahara and South Asia, contribution of $F_{toa}$, and GMS, & (**b**) the difference in $F_{toa}$ and its components. The change in OLR is further decomposed into changes due to clear sky OLR, changes in cloud area fraction, the longwave cloud absorption, and non-linear term (see Appendix A4). (**c**) & (**d**) illustrates the relation between clear sky OLR, column-integrated water vapor and surface temperature, respectively. JJA climatology (1981-2020) from ERA-5 are considered for this analysis. The regions selected for the analysis are shown in the inset map with grey shading: $0°E$-$30°E$ and $15°N$-$30°N$; land-only grid points (Sahara) and $70°E$-$100°E$ and $15°N$-$30°N$; all grid points (South Asia).

The precipitation rate and $F_{toa}$ over the RETRO Sahara change substantially in the first summer after the rotation is reversed (Supplementary Fig. 7a & 7b). The surface albedo is nearly the same as that in CTL (Supplementary Fig. 7c). The increase in $F_{toa}$ and precipitation is due to the clear sky radiative effects of water vapor (Supplementary Fig. 8a & 8b). A closer look at the daily evolution of $F_{toa}$ during the first year of simulations highlight the important role played by OLR in the transition of $F_{toa}$ from negative to positive values (Supplementary Fig. 9a, 9b,& 9c). With the reversal in Earth's rotation, the large-scale dynamics change. This leads to the advection of moisture into the Sahara. The radiative effect of water vapor leads to an



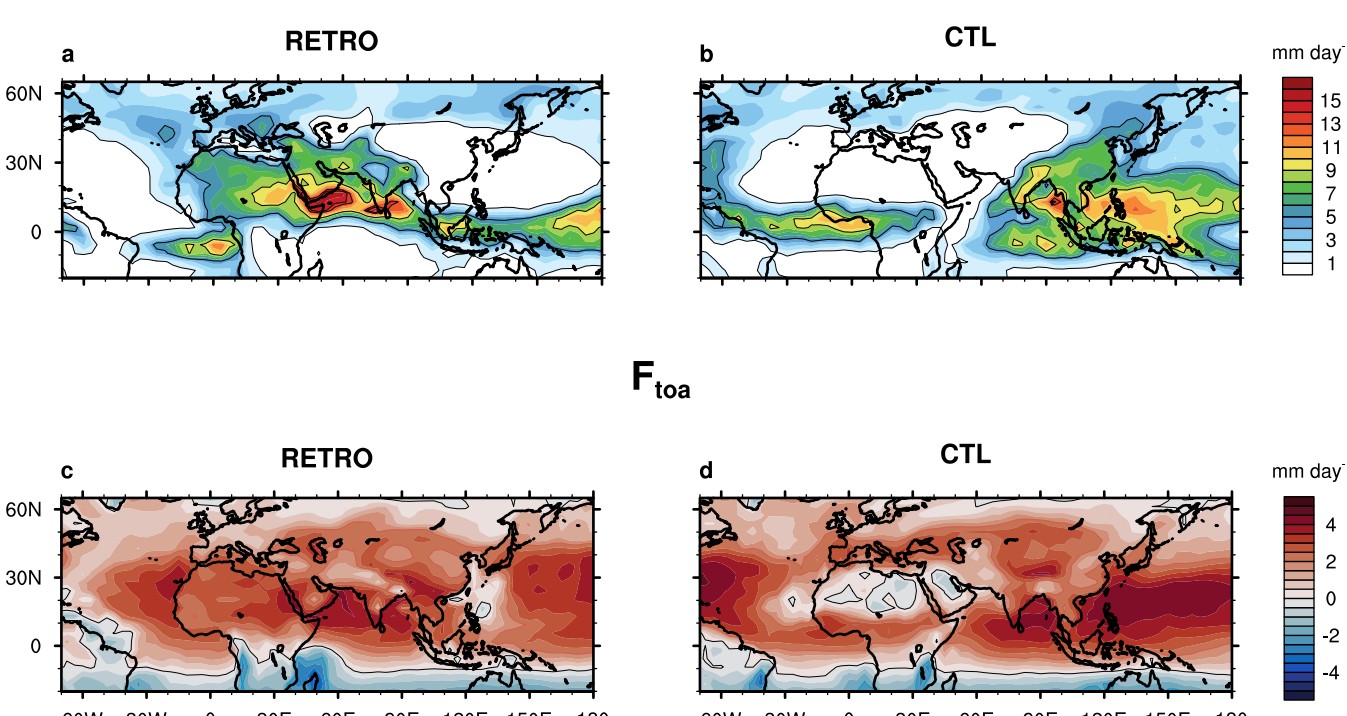

**Figure 3. Summer mean precipitation, and F$_{toa}$.** The spatial plot depicts the Jun-Jul-Aug (JJA) climatology of precipitation rate (**a** & **b**), and net radiation at the top of the atmosphere (F$_{toa}$) (**c** & **d**). (**a**, & **c**) are for RETRO (retrograde Earth) simulation, and (**b**, & **d**) are for the CTL (control) simulation. Climatology over the last 100 years of the simulation is considered.

increase in F$_{toa}$. The atmospheric circulation responds to these changes in F$_{toa}$ by advecting additional MSE out of the domain. This leads to an increased moisture flux into the domain. Thus, a feedback of water vapor radiative effect on circulation is established. Clouds also form in this process and modulate the local F$_{toa}$ through their radiative effects. In CTL, adiabatic subsidence prevailed over the Sahara, which is now absent. This also aids in the formation of clouds. These feedbacks and

processes lead to the development of a strong monsoon over the RETRO Sahara despite its high albedo. Vegetation grows over the course of a few decades, reducing albedo (Supplementary Fig. 7c). This is, however, a slow feedback and contributes approximately 35% to the final change in precipitation.

## 3    Discussion and Conclusions

Monsoons and deserts are contrasting climates, that are believed to be shaped by distinct dynamics (Sooraj et al., 2021). These

differences also manifest in the top-of-atmosphere radiation budget (Charney, 1975). The large-scale atmospheric dynamics balances the TOA radiation budget through the advection of energy. Thus, both climates can be described through a common



# Column integrated water vapor

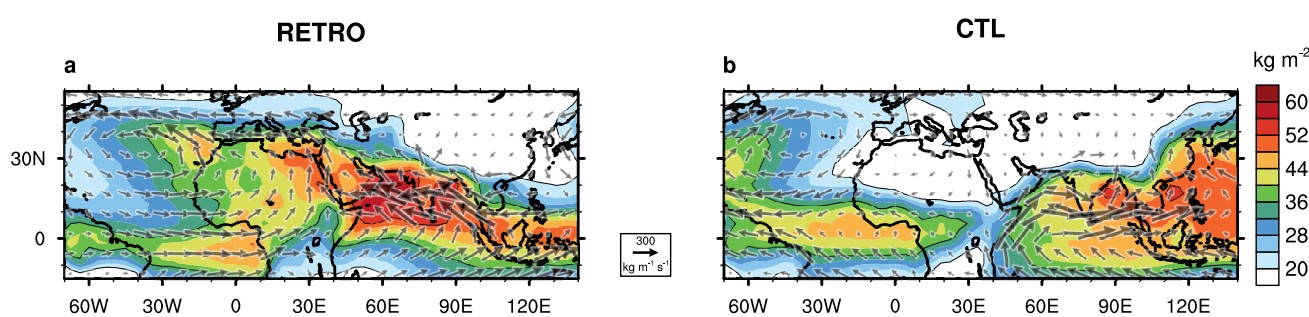

# Omega (500 hPa)

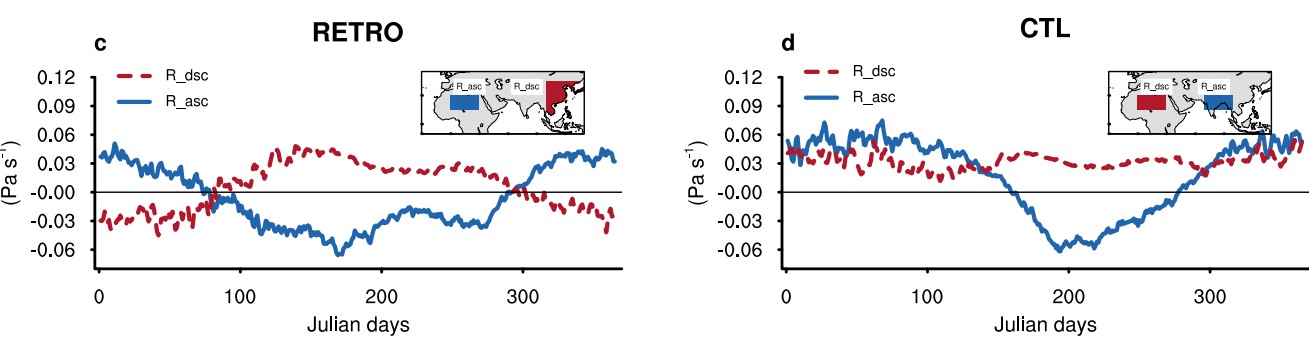

**Figure 4. Comparison of water vapor flux between CTL and RETRO.** The spatial map of column-integrated water vapor (in colors) and the vertically integrated moisture flux (vectors) for the (**a**) RETRO and (**b**) CTL. The JJA climatology over the last 100 years of each simulation is considered. The time evolution of vertical pressure velocity ($\omega$) at 500 hPa for the (**c**) RETRO and (**d**) CTL. The solid blue line and the dashed red line are area weighted averages of $\omega_{500}$ over the regions of corresponding colors shown in the inset map.

radiation-circulation framework. Traditionally, the differences in the TOA radiation budget between monsoons and deserts have been attributed to the differences in surface albedo (Charney, 1975). Feedbacks from surface albedo on local radiative cooling is considered to be important for the differences in TOA radiation budget between monsoons and deserts. However, a

delineation of the sequence of events that produces TOA radiation budget contrast has been missing. Our results suggest that radiative effect of water vapor and cloud cover contribute nearly equally to the differences in the TOA radiation budget. Before the onset of the monsoon, the radiative effect of water vapor exerts the greatest influence on the TOA radiation budget. As the monsoon progresses and clouds begin to form, cloud radiative effects further intensify, amplifying the contrast in the TOA radiation budget between monsoon and desert regions.

Using a RETRO simulation and the energetics framework we present a scenario where monsoon exists over a region of high surface albedo. With this simulation and our diagnostics, we unravel for the first time, the processes and sequence of events



that produce monsoon-desert TOA radiation budget contrast. During the first summer after a reversal in the rotation of Earth has taken effect in RETRO, the large-scale circulation transports moisture into the Sahara. This has an instantaneous effect on the outgoing longwave radiation and, therefore, on the TOA radiation budget. Atmospheric circulation adjusts in response to

changes in the TOA radiation budget, further amplifying the moisture transport into the region. Thus, the water vapor radiative effect feeds back onto the large-scale circulation. We suggest that this radiative feedback of water vapor on circulation plays a crucial role in the formation of monsoons. As convection is initiated, cloud radiative effects further modulate the local TOA radiation budget, eventually leading to the development of a monsoon over the Sahara despite its high albedo. The subsequent growth of vegetation strengthens the monsoon over time. Vegetation-albedo feedback is a slow process that operates over a

few decades contributing to 35% of the precipitation changes. The water vapor/clouds – OLR – circulation feedback is a fast process that accounts for about 65% of the increase in precipitation. As monsoon strengthens over the Sahara in RETRO, subsidence is induced over East Asia inhibiting convection and promoting arid conditions. Since East Asia continues to receive precipitation during the boreal winter, vegetation exists. The changes in circulation ensure an arid climate only during the boreal summer. This does not, however, lead to a reduction in the TOA radiation budget over East Asia in RETRO. This highlights

that desertification is primarily driven by dynamics, with the surface albedo feedbacks playing a secondary role.

Our results have implications for the seasonal development of monsoons in the modern climate. The historical perspective of monsoons has been synonymous with land-sea thermal contrast (Halley, 1753; Hadley, 1735; Lau and Li, 1984; Meehl, 1992). The modern view of monsoons is that of an energetically driven meridional movement of the interhemispheric convergence zone (ITCZ) (Gadgil, 2018; Hill, 2019). Both the historical and modern perspectives of monsoons do not consider feedbacks

internal to the monsoon system. The onset and progression of monsoons is akin to the development of a monsoon over the Sahara in RETRO. During onset, changes in large-scale circulation occur that advect large amounts of moisture into the monsoon domain. This triggers the radiation-circulation feedback through clear sky radiative effects of water vapor. Conceptual models, that are extensively used for monsoon research, need to consider this feedback to comprehensively investigate monsoon dynamics. Once the monsoon is established, the associated diabatic heating strengthens the atmospheric subsidence to the west

of the monsoon region (as described by the Rodwell-Hoskins mechanism (Rodwell and Hoskins, 1996)), creating an arid climate. Previous studies have proposed that the vegetation-albedo feedback at the surface further amplifies this subsidence, contributing to the net difference in the TOA radiation budget between monsoon regions and deserts. However, our findings highlight the critical role of water vapor radiative effect on circulation. In summary, our results underscore the importance of water vapor in establishing and reinforcing a radiation-circulation feedback, thereby accentuating the contrast in the TOA

radiation budget between monsoon regions and deserts.

*Code availability.* The codes are available from the corresponding author on request.



*Data availability.* The RETRO and CTL simulations used in this article can be downloaded from the World Data Center for Climate (DKRZ) archives (https://www.wdc-climate.de/ui/entry?acronym=DKRZ_LTA_110_ds00001). The ERA5 data are available at https://cds.climate. copernicus.eu/cdsapp#!/dataset/reanalysis-era5-single-levels?tab=form. The CERES EBAF data were obtained from https://ceres-tool.larc.
nasa.gov

## Appendix A: Data and Methods

### A1   Model and experimental setup

The CTL and RETRO simulations were carried out using the MPI-ESM v1.2 (Mauritsen et al., 2019). This model consists of the atmospheric general circulation model ECHAM 6.3.02 (Stevens et al., 2013) (with some bug fixes) and the land model
JSBACH 3.10 (Reick et al., 2013) with a dynamic vegetation module. The ocean general circulation model is the MPIOM 1.6.2p3 (Jungclaus et al., 2013), with the marine biogeochemistry model HAMOCC (Ilyina et al., 2013; Paulsen et al., 2017) and a dynamic-thermodynamic sea-ice model (Notz et al., 2013). The model was run at coarse resolution: $3.75° \times 3.75°$ for the atmosphere (T31) and a nominal resolution of $3°$ for the ocean.

The CTL simulation is the pre-industrial setup and follows the CMIP5 protocol. In RETRO, the sign of the Coriolis parameter
and the direction of the Sun's diurnal march were reversed. The boundary conditions in RETRO are the same as that in CTL. Both simulations were initialized from CTL and run for 6990 years till steady state is reached. The climatology of the last 100 years is used for all the analysis. The precipitation and $F_{toa}$ in CTL display the monsoon-desert asymmetry (Fig. 3b & d) consistent with modern observations. The daily data used in Fig. 4b & 4c are taken from the climatology of the last 30 years of the simulation. Daily data from the transient model spin-ups are used to understand the transition of the Sahara (Supplementary
Fig. 7, 8, 9). Further details about the model and the simulations can be found in Mikolajewicz et al. (Mikolajewicz et al., 2018).

### A2   Diagnosis using energetics

The energetics framework relates the moisture converged into a region to the net energy input into the atmosphere ($Q_{div}$) and gross moist stability (GMS). $Q_{div}$ is the total of the TOA radiation budget ($F_{toa}$) and $F_{sfc}$. $F_{sfc}$ is the summation of all surface energy flux entering the atmosphere. Since, $F_{sfc}$ is insignificant in the regions of interest, $Q_{div}$ is predominantly equal to
$F_{toa}$. This definition of GMS includes all advection terms (Jalihal et al., 2019a, b, 2020). This is particularly relevant because previous literature has shown that horizontal advection terms are important for deserts as well as monsoon domains (Tyrlis et al., 2013; Cherchi et al., 2014; Jalihal et al., 2019a). Several models for tropical circulation based on energetics exist in literature. However, they assume a simple baroclinic structure of the vertical velocity. This does not represent accurately the omega profile over deserts (Supplementary Fig. 3a). Hence, we do not use these models for tropical circulation and use precipitation minus
evaporation (P-E) instead. P-E is nearly zero over deserts, and is positive over monsoons. This can be used as a criterium to diagnose the two climates (Neelin and Held, 1987). Hence, we use the following version:

$$P - E = \frac{Q_{\text{div}}}{GMS} \approx \frac{F_{\text{toa}}}{GMS}$$



where, $P$ and $E$ are the precipitation rate and evaporation rate, respectively. Since, $P - E$ is in the units of mm day$^{-1}$, F$_{toa}$ and F$_{sfc}$ are also converted to mm day$^{-1}$ (taking the latent heat of vaporization of water as 2.501x10$^6$ J kg$^{-1}$ we get 1 mm day$^{-1}$ = 28.95 W m$^{-2}$).

The difference in $P - E$ between RETRO and CTL can be written as:

$$\underbrace{\Delta(P-E)}_{\text{Change in P-E}} = \underbrace{\frac{\Delta F_{\text{toa}}}{1 + \frac{\Delta GMS}{GMS}}(P-E)}_{\text{Contribution from change in F}_{\text{TOA}}} + \underbrace{\frac{\Delta GMS}{1 + \frac{\Delta GMS}{GMS}}(P-E)}_{\text{Contribution from change in GMS}}$$

where, $\Delta$ is the difference between RETRO and CTL. The absolute variables (without $\Delta$) are from the reference climate (CTL).

### A3    Net top of the atmosphere radiation budget (F$_{toa}$)

The net radiative energy flux at the top of the atmosphere (in mm day$^{-1}$) is given by:

$$F_{toa} = S(1 - \alpha) - OLR$$

where, $S$ is the incoming solar radiation, $\alpha$ is the shortwave reflectivity at the top of the atmosphere, $OLR$ is the outgoing
longwave radiation. Positive sign represent addition of energy into the atmosphere. $F_{sfc}$ is the total of all surface energy fluxes:

$$F_{sfc} = SHF + LHF + NSW_{sfc} + NLW_{sfc}$$

where, $SHF$, and $LHF$ are the surface sensible and latent heat fluxes. $NSW_{sfc}$ is the net surface shortwave radiation, and $NLW_{sfc}$ is the net surface longwave radiation. The difference in F$_{toa}$ between RETRO and CTL, assuming insolation does not change, is:

$$\Delta F_{toa} = -\Delta\alpha S - \Delta OLR$$

Here, $\Delta$ is the difference between RETRO and CTL. Since the regions of interest are land grids (Sahara, South East and East Asia), F$_{sfc}$ is negligible due to low thermal storage of land. Moreover, F$_{sfc}$ is also nearly zero over the Bay of Bengal (Ramesh and Boos, 2022) (particularly, the north Bay of Bengal which forms a part of the South Asian monsoon domain considered in
this study). Hence, we use F$_{toa}$ over the regions of interest.

### A4    Decomposition of outgoing longwave radiation (OLR).

The area averaged OLR over a domain can be expressed as:

$$OLR = OLR_{\text{clr}} \times (1 - A_{\text{cld}}) + OLR_{\text{cld}} \times A_{\text{cld}}$$





$OLR_{clr}$ and $OLR_{cld}$ are the clear sky and cloudy sky longwave emission at the top of the atmosphere. $A_{cld}$ is the cloud area fraction over the whole domain. This equation can be rearranged as follows:

$$OLR = OLR_{clr} + (OLR_{cld} - OLR_{clr}) \times A_{cld}$$

Taking the difference of the above equation between RETRO and CTL gives us:

$$\Delta OLR = \underbrace{\Delta OLR_{clr}}_{\text{Change in clearsky OLR}} + \underbrace{(OLR_{cld} - OLR_{clr}) \times \Delta A_{cld}}_{\text{Change in cloud area fraction}} + \underbrace{A_{cld} \times \Delta (OLR_{cld} - OLR_{clr})}_{\text{Longwave cloud absorption}} +$$

$$\underbrace{\Delta (OLR_{cld} - OLR_{clr}) \times \Delta A_{cld}}_{\text{Non-Linear term}}$$

Here, $\Delta$ is the difference between RETRO and CTL. In the above equation, the variables with absolute values are taken from the reference climate - CTL.

*Author contributions.* C.J., and U.M. analysed and interpreted the GCM output. C.J. wrote the manuscript with input from U.M. All authors reviewed the manuscript.

*Competing interests.* The authors declare that they have no competing financial interests.

*Acknowledgements.* We thank B. Stevens and J. Srinivasan for their helpful comments and discussions. We are grateful to J. Marotzke, M-L Kapsch, and C. Schannwell for their constructive feedback on the manuscript. C. Jalihal is funded by the Alexander von Humboldt foundation, project number : IND 1222628 HFST-P. We gratefully acknowledge the Deutsches Klimarechenzentrum (DKRZ) for providing the computational resources.



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
