# Peer review of "Energetics of monsoons and deserts: role of surface albedo vs water vapor feedback"

_EGUsphere, 2025_

## Author Comment (AC1)

**Response to referee 1's comments "Energetics of monsoons and deserts: role of surface albedo vs water vapor feedback"**

Chetankumar Jalihal1,2 and Uwe Mikolajewicz2

The authors thank the referees for their time and inputs. We have provided a point-by-point response to each of their comments.

General comment: In this manuscript, the authors present a compelling argument that the top-of-the-atmosphere (TOA) radiation budget contrast between monsoon and desert regions is primarily driven by water vapour feedback, with surface albedo playing only a secondary role. This challenges the classical Charney (1975, https://doi.org/10.1002/qj.49710142802) hypothesis, which emphasises albedo-driven desertification feedbacks. The study employs a combination of theoretical reasoning and a novel climate model experiment (RETRO, in which Earth's rotation is reversed) to support its claims. While the hypothesis is intriguing and potentially significant for understanding monsoon-desert radiative dynamics, I have some serious concerns regarding the experimental design and interpretation of results.

**1. Comment: Major Concern: Limitations of the RETRO Experiment**

The central issue with this study lies in its reliance on the RETRO experiment to "confirm" the hypothesis. While reversing Earth's rotation is a creative way to alter large-scale climate asymmetries, it is not an appropriate experimental framework for isolating the specific roles of water vapour versus surface albedo in TOA radiation budgets. My concerns are as follows:

**Fundamental Alteration of Planetary Dynamics**

Reversing Earth's rotation drastically changes the Coriolis force, jet stream pathways, ocean circulation, and storm tracks. These modifications introduce confounding dynamical effects that are unrelated to water vapour's radiative role. The resulting climate (e.g., a Sahara monsoon and Southeast Asian desert in the RETRO simulation) is influenced not just by humidity and radiation but also by completely reconfigured atmospheric and oceanic circulations. Thus, attributing the TOA budget differences solely to water vapour is problematic.

**Reply:** It is true that reversing the direction of planetary rotation alters large-scale atmospheric dynamics. Our central proposition is that this shift acts as an initial trigger for the onset of monsoon conditions over the Sahara. The subsequent radiative effects from water vapor and clouds then feed back into the circulation, further amplifying it (see Lines 132–137 and 151–153 of the main text). This feedback emerges organically in the RETRO simulation. Notably,

<sup>1Max Planck Institute for Meteorology, Hamburg

<sup>2Department of Climate Change, Indian Institute of Technology Hyderabad, India

even in the first year of the simulation—when surface albedo remains high—the large-scale circulation modulates the local radiative budget over the Sahara. During this period,  $F_{toa}$  becomes positive and monsoon-like conditions develop. This provides compelling evidence that surface albedo is not the primary limiting factor for monsoon onset. Thus, this simulation reveals how radiative feedbacks from water vapor and clouds reinforce the dynamical changes that occur upon reversing the rotation. It also supports the argument surface properties are not the primary limiting factor.

We address two key aspects in this context:

- Climatological differences in Ftoa and their link to low-level mass convergence Neelin and Held (1987) demonstrate that, under steady-state conditions, low-level mass convergence is proportional to Ftoa (over land, over oceans one must consider the surface energy fluxes as well), with the gross moist stability (GMS) serving as the proportionality constant. Equation 2.12 in their paper encapsulates this relationship. When Ftoa is near zero or negative, it implies minimal or divergent low-level flow—exactly the condition observed over the Sahara. Thus, spatial variations in low-level circulation can be diagnosed through Ftoa. In our analysis, differences in Ftoa between monsoon and desert regions point to a dominant role played by water vapor and cloud radiative effects.
- Seasonal evolution of Ftoa and its relationship to large-scale circulation Before the onset of the South Asian monsoon, the absorbed shortwave radiation (ASR) and OLR over South Asia are similar to those over the Sahara (see Figure ??). The main difference arises in the evolution of OLR. Initially, rising water vapor levels increase Ftoa, which enhances low-level convergence (from Neelin and Held (1987)). This, in turn, draws in more moisture, further increasing Ftoa and reinforcing the circulation. Cloud radiative effects lag behind those of water vapor by a few days, but once clouds begin to form, they contribute additional radiative forcing that further modulates Ftoa and the low-level convergence.

Figure 1. Seasonal cycle of  $F_{toa}$  and its components. The time series of (a)  $F_{toa}$ , (b) Net shortwave at the top of atmosphere, and (c) outgoing longwave radiation from ERA-5 climatology (based on 1981-2020). The solid line represents the area average over the domain (70°E-105°E and 15°N-30°N), while the dashed line shows the area average over the domain (0°E-30°E and 15°N-30°N).

2. Comment: Lack of a Clean Sensitivity Test A more robust approach would involve directly perturbing water vapour concentrations (e.g., through a "dry world" vs. "moist world" experiment) while keeping Earth's rotation unchanged. Alternatively, radiative kernel analysis could quantitatively separate the contributions of water vapour, clouds, and surface albedo to the TOA budget. Please refer to Soden et al. (2008, https://doi.org/10.1175/2007JCLI2110.1) for further details.

**Reply:** We thank the referee's suggestion. Conducting "dry world" versus "moist world" experiments is indeed an interesting idea. However, implementing a fully dry atmosphere in a comprehensive Earth System Model (ESM) would introduce unintended consequences. Radiative kernels, which are commonly used for such diagnostics, are linearized around a specific base climate. Te RETRO simulation represents a substantial shift in climate. Radiative kernels suitable for RETRO do not exist.

Our primary objective is to diagnose the dominant factors contributing to the TOA energy budget differences between monsoons and deserts. For this purpose, we find that an offline radiative transfer model offers a more controlled and transparent framework. We use the Climlab implementation of RRTMG (Rose, 2018), which is consistent with the radiative scheme employed in our MPI-ESM simulations. We prescribe the thermodynamic profiles and aerosol properties are prescribed.

The model reproduces clear-sky and all-sky OLR with errors below 1%. Errors in ASR are slightly higher (2–3%), primarily due to the absence of cloud optical properties in the standard model output, which are needed for accurate ASR calculations. To isolate the contributions of individual components—such as temperature, water vapor, and clouds—we apply the Partial Radiative Perturbation (PRP) technique (Box, 2002), which allows us to quantify their respective impacts on TOA fluxes.

As shown in Figure ??, the reflected shortwave radiation at TOA is nearly identical over the Sahara and South Asia during JJA, with a difference of approximately 25 W m-2 in both the CERES and ERA5 datasets. The surface albedo of the Sahara is nearly the same as the TOA albedo over a cloud covered South Asia. The dominant contributor to the TOA energy budget difference between these regions is the OLR (about 90 W m-2). The figure below (Figure 2) shows the individual contributions of clouds, water vapor, temperature (surface and atmospheric), and aerosols to the difference in all-sky OLR between South Asia and the Sahara. Clouds and water vapor exert the strongest influence. Particularly, during the onset of the monsoon, the radiative impact of water vapor increases first, followed by a more pronounced contribution from clouds. In the RETRO simulation,

3.

The study does not account for dust aerosols, which are prevalent over deserts and significantly influence both shortwave (albedo) and longwave (trapping) radiation (Osborne et al., 2011, QJRMS, https://doi.org/10.1002/qj.771). The role of cloud feedbacks, while briefly mentioned, is not rigorously disentangled from water vapour effects. Since clouds co-vary

**S.Asia - Sahara**

**Figure 2. Decomposing longwave emission** The time series of all-sky OLR difference between South Asia and the Sahara based on offline RRTMG calculations. IThe black line represents the total all-sky OLR difference. Individual contributions from clouds, water vapor, temperature (surface and atmosphere), and aerosols are shown in grey, blue, red, and green, respectively.

with humidity, their radiative impact could also explain part of the TOA contrast. *Comment: Overlooked Factors: Dust Aerosols and Clouds*

The study does not account for dust aerosols, which are prevalent over deserts and significantly influence both shortwave (albedo) and longwave (trapping) radiation (Osborne et al., 2011, QJRMS, https://doi.org/10.1002/qj.771). The role of cloud feedbacks, while briefly mentioned, is not rigorously disentangled from water vapour effects. Since clouds co-vary with humidity, their radiative impact could also explain part of the TOA contrast.

**Reply:** Thank you for your the comment. We have now included in our analysis an assessment of the impact of aerosols. We find that aerosols play a minor role during JJA. Their influence on all-sky OLR is relatively higher during

**RETRO - CTL**

**Figure 3. Decomposing longwave emission in RETRO Sahara** The time series of all-sky OLR difference between RETRO and CTL over the Sahara based on offline RRTMG calculations. IThe black line represents the total all-sky OLR difference. Individual contributions from clouds, water vapor, temperature (surface and atmosphere), and aerosols are shown in grey, blue, red, and green, respectively.

the pre-monsoon period. Since, pre-industrial aerosols are prescribed to the RETRO, their contribution to the TOA budget does not change. Aerosols contribute about 10 W  $m_{-2}$  to the ASR at TOA (not shown) and thus, play only a minor role. Cloud radiative effects lag the water vapor radiative effect (Figure 2 & 3). This lag is not as pronounced in RETRO.

**References**

- Box, M. A.: Radiative perturbation theory: a review, Environmental Modelling & Software, 17, 95-106, 2002.
- Neelin, J. D. and Held, I. M.: Modeling tropical convergence based on the moist static energy budget, Monthly Weather Review, 115, 3–12, 1987.
- Rose, B. E.: CLIMLAB: a Python toolkit for interactive, process-oriented climate modeling., J. Open Source Softw., 3, 659, 2018.

---

## Author Comment (AC3)

**Response to referee 1's comments "Energetics of monsoons and deserts: role of surface albedo vs water vapor feedback"**

Chetankumar Jalihal1,2 and Uwe Mikolajewicz2

The authors thank the referees for their time and inputs. We have provided a point-by-point response to each of their comments.

General comment: In this manuscript, the authors present a compelling argument that the top-of-the-atmosphere (TOA) radiation budget contrast between monsoon and desert regions is primarily driven by water vapour feedback, with surface albedo playing only a secondary role. This challenges the classical Charney (1975, https://doi.org/10.1002/qj.49710142802) hypothesis, which emphasises albedo-driven desertification feedbacks. The study employs a combination of theoretical reasoning and a novel climate model experiment (RETRO, in which Earth's rotation is reversed) to support its claims. While the hypothesis is intriguing and potentially significant for understanding monsoon-desert radiative dynamics, I have some serious concerns regarding the experimental design and interpretation of results.

**1. Comment: Major Concern: Limitations of the RETRO Experiment**

The central issue with this study lies in its reliance on the RETRO experiment to "confirm" the hypothesis. While reversing Earth's rotation is a creative way to alter large-scale climate asymmetries, it is not an appropriate experimental framework for isolating the specific roles of water vapour versus surface albedo in TOA radiation budgets. My concerns are as follows:

**Fundamental Alteration of Planetary Dynamics**

Reversing Earth's rotation drastically changes the Coriolis force, jet stream pathways, ocean circulation, and storm tracks. These modifications introduce confounding dynamical effects that are unrelated to water vapour's radiative role. The resulting climate (e.g., a Sahara monsoon and Southeast Asian desert in the RETRO simulation) is influenced not just by humidity and radiation but also by completely reconfigured atmospheric and oceanic circulations. Thus, attributing the TOA budget differences solely to water vapour is problematic.

**Reply:** It is true that reversing the direction of planetary rotation alters large-scale atmospheric dynamics. Our central proposition is that this shift acts as an initial trigger for the onset of monsoon over the Sahara. The subsequent radiative effects from water vapor and clouds then feed back into the circulation, further amplifying it (as mentioned in Lines 132–137 and 151–153 of the main text). This feedback emerges organically in the RETRO simulation. Notably,

<sup>1Max Planck Institute for Meteorology, Hamburg

<sup>2Department of Climate Change, Indian Institute of Technology Hyderabad, India

even in the first year of the simulation—when surface albedo remains high—the large-scale circulation modulates the local radiative budget over the Sahara by advecting moisture and subsequently through the formation of clouds. During this period,  $F_{toa}$  becomes positive and monsoon-like conditions develop. This provides compelling evidence that surface albedo is not the primary limiting factor for monsoon. Thus, this simulation reveals how radiative feedbacks from water vapor and clouds reinforce the dynamical changes that occur upon reversing the rotation. It also supports the argument that surface properties are not the primary limiting factor.

We address two key aspects in this context:

- Climatological differences in Ftoa and their link to low-level mass convergence Neelin and Held (1987) demonstrate that, under steady-state conditions, low-level mass convergence is proportional to Ftoa (over land, over oceans one must consider the surface energy fluxes as well), with the gross moist stability (GMS) serving as the proportionality constant. Equation 2.12 in their paper encapsulates this relationship. When Ftoa is near zero or negative, it implies minimal or divergent low-level flow—exactly the condition observed over the Sahara. Thus, spatial variations in low-level circulation can be diagnosed through Ftoa. In our analysis, differences in Ftoa between monsoon and desert regions point to a dominant role played by water vapor and cloud radiative effects.
- Seasonal evolution of Ftoa and its relationship to large-scale circulation Before the onset of the South Asian monsoon, the absorbed shortwave radiation (ASR) and OLR over South Asia are similar to those over the Sahara (see Figure 1). The main difference arises in the evolution of OLR. Initially, rising water vapor levels increase Ftoa, which enhances low-level convergence (from Neelin and Held (1987)). This, in turn, draws in more moisture, further increasing Ftoa and reinforcing the circulation. Cloud radiative effects lag behind those of water vapor by a few days, but once clouds begin to form, they contribute additional radiative forcing that further modulates Ftoa and the low-level convergence.

Figure 1. Seasonal cycle of  $F_{toa}$  and its components. The time series of (a)  $F_{toa}$ , (b) Net shortwave at the top of atmosphere, and (c) outgoing longwave radiation from ERA-5 climatology (based on 1981-2020). The solid line represents the area average over the domain (70°E-105°E and 15°N-30°N), while the dashed line shows the area average over the domain (0°E-30°E and 15°N-30°N).

2. Comment: Lack of a Clean Sensitivity Test A more robust approach would involve directly perturbing water vapour concentrations (e.g., through a "dry world" vs. "moist world" experiment) while keeping Earth's rotation unchanged. Alternatively, radiative kernel analysis could quantitatively separate the contributions of water vapour, clouds, and surface albedo to the TOA budget. Please refer to Soden et al. (2008, https://doi.org/10.1175/2007JCLI2110.1) for further details.

**Reply:** We thank the referee for the suggestion. Conducting "dry world" versus "moist world" experiments is indeed an interesting idea. However, implementing a fully dry atmosphere in a comprehensive Earth System Model (ESM) would introduce unintended consequences. In an interesting study, Byrne et al. (2018) demonstrated the impact of clouds and water vapor radiative effects on monsoons using radiation-locking simulations in an axisymmetric slab-ocean model. In contrast, our approach focuses on comparing distinct climate regimes (monsoon vs. desert, or RETRO vs. control) to isolate the roles of clouds and water vapor without interfering with the ESM's underlying physics.

Radiative kernels, which are commonly used for such diagnostics, are linearized around a specific base climate. The RETRO simulation represents a substantial shift in climate. Radiative kernels suitable for RETRO do not exist.

Our primary objective is to diagnose the dominant factors contributing to the TOA energy budget differences between monsoons and deserts. For this purpose, we find that an offline radiative transfer model offers a more controlled and transparent framework. We use the Climlab implementation of RRTMG (Rose, 2018) (MPI-ESM uses RRTMG). We prescribe the thermodynamic profiles and aerosol properties.

The model reproduces clear-sky and all-sky OLR with errors below 1%. Errors in ASR are slightly higher (2–3%), primarily due to the absence of cloud optical properties in the standard model output, which are needed for accurate ASR calculations. To isolate the contributions of individual components—such as temperature, water vapor, and clouds—we apply the Partial Radiative Perturbation (PRP) technique (Box, 2002), which allows us to quantify their respective impacts on TOA fluxes.

As shown in Figure 1, the reflected shortwave radiation at TOA over the Sahara and South Asia during JJA have a difference of approximately 25 W m-2 (1 mm day-1 = 28.98 W m-2) (this is consistent in both the CERES and ERA5 datasets). The dominant contributor to the TOA energy budget difference between these regions is the OLR (about 90 W m-2). Hence, we further examine the impact of various factors on OLR. The figure below (Figure 2) shows the individual contributions of clouds, water vapor, temperature (surface and atmospheric), and aerosols to the difference in all-sky OLR between South Asia and the Sahara. Clouds and water vapor exert the strongest influence. Particularly, during the onset of the monsoon, the radiative impact of water vapor increases first, followed by a more pronounced contribution from clouds. This result is less pronounced in the RETRO Sahara simulation (Figure 3), primarily due to the domain selected. Pre-monsoon rainfall in the RETRO Sahara, especially in the northernmost part of the domain, is largely influenced by transient mid-latitude storms (In this animation the propagation of mid-latitude storms over the northern Sahara in RETRO can be seen - link). These storms are driven by baroclinic instabilities and lead to rapid cloud development through frontal lifting. Consequently, changes in water vapor are closely tied to cloud evolution. As a result, distinguish-

**S.Asia - Sahara**

**Figure 2. Decomposing longwave emission** The time series of all-sky OLR difference between South Asia and the Sahara based on offline RRTMG calculations. The black line represents the total all-sky OLR difference. Individual contributions from clouds, water vapor, temperature (surface and atmosphere), and aerosols are shown in grey, blue, red, and green, respectively.

ing the lead–lag relationship between water vapor and cloud radiative effects on OLR during non-monsoonal months is challenging at daily resolution. In contrast, monsoon onset is governed by large-scale dynamics, where moisture is gradually advected from the oceans over a relatively stable region. Once sufficient moisture accumulates to destabilize the atmosphere, convection initiates and clouds form. Therefore, during monsoon onset, water vapor radiative effects precede those of clouds. Selecting a region within the RETRO Sahara that is less affected by baroclinic instabilities allows for a clearer representation of monsoon onset and the sequential radiative impacts of water vapor and clouds. We choose the region depicted in the inset map in Figure 4. This figure demonstrates that our results do not change when we choose a different domain. All the changes in moisture convergence (P-E) is related to changes in  $F_{toa}$  (Figure 4a). Decomposing  $F_{toa}$  into its components indicates that OLR is the dominant factor (Figure 4b). Our analysis with the

RRTMG reveals that water vapor radiative effects dominate initially and subsequently the cloud radiative effects take over (Figure 5).

**Figure 3. Decomposing longwave emission in RETRO Sahara** The time series of all-sky OLR difference between RETRO and CTL over the Sahara based on offline RRTMG calculations. IThe black line represents the total all-sky OLR difference. Individual contributions from clouds, water vapor, temperature (surface and atmosphere), and aerosols are shown in grey, blue, red, and green, respectively.

**JJA Climatology**

Figure 4. Diagnosis of change in moisture convergence during the first year of simulation. Bar graph of (a) change in (Precipitation - Evaporation; P-E), contribution of  $F_{toa}$ , and GMS, & (b) the change in  $F_{toa}$  and its components (see methods). The changes between Jun-Jul-Aug average of the first year of simulation from the RETRO and Jun-Jul-Aug climatology from the CTL over the domain  $(20^{\circ}\text{E}-50^{\circ}\text{E} \text{ and } 10^{\circ}\text{N}-25^{\circ}\text{N})$  (land only grids) shown in grey shading in the inset map), is considered for this analysis. The change in OLR is further decomposed into changes due to clear sky OLR, changes in cloud area fraction, the longwave cloud absorption, and non-linear term (see methods).

3. Comment: Overlooked Factors: Dust Aerosols and Clouds The study does not account for dust aerosols, which are prevalent over deserts and significantly influence both shortwave (albedo) and longwave (trapping) radiation (Osborne et al., 2011, QJRMS, https://doi.org/10.1002/qj.771). The role of cloud feedbacks, while briefly mentioned, is not rigorously disentangled from water vapour effects. Since clouds co-vary with humidity, their radiative impact could also explain part of the TOA contrast.

**Reply:** Thank you for the comment. We have now included in our analysis an assessment of the impact of aerosols. We find that aerosols play a minor role during JJA. Their influence on all-sky OLR is relatively higher during the premonsoon period. Since, pre-industrial aerosols are prescribed to the RETRO, their contribution to the TOA budget does not change. Aerosols contribute about 10 W m-2 to the ASR at TOA (not shown) and thus, play only a minor role.

The radiative effects of clouds on monsoons has been examined in previous studies (Berry and Mace, 2014; Li et al., 2017; Byrne and Zanna, 2020; Stephens et al., 2024; Wang et al., 2020). In contrast this study highlights the role of water vapor radiative effects. To our knowledge, Byrne et al. (2018) is the only other study to have examined the impact of

**RETRO - CTL**

Figure 5. Diagnosis of change in moisture convergence during the first year of simulation. Bar graph of (a) change in (Precipitation - Evaporation; P-E), contribution of  $F_{toa}$ , and GMS, & (b) the change in  $F_{toa}$  and its components (see methods). The changes between Jun-Jul-Aug average of the first year of simulation from the RETRO and Jun-Jul-Aug climatology from the CTL over the domain ( $20^{\circ}E-50^{\circ}E$  and  $10^{\circ}N-25^{\circ}N$  (land only grids) shown in grey shading in the inset map), is considered for this analysis. The change in OLR is further decomposed into changes due to clear sky OLR, changes in cloud area fraction, the longwave cloud absorption, and non-linear term (see methods).

water vapor radiative effects on monsoon, albeit in a simplified and idealized model. Hence, our focus has been on the radiative effects of water vapor. In the new manuscript we will enhance the discussion about aerosols and clouds.

**References**

- Berry, E. and Mace, G. G.: Cloud properties and radiative effects of the Asian summer monsoon derived from A-Train data, Journal of Geophysical Research: Atmospheres, 119, 9492–9508, 2014.
- Box, M. A.: Radiative perturbation theory: a review, Environmental Modelling & Software, 17, 95–106, 2002.
- Byrne, M. P. and Zanna, L.: Radiative effects of clouds and water vapor on an axisymmetric monsoon, Journal of Climate, 33, 8789–8811, 2020.
- Byrne, M. P., Pendergrass, A. G., Rapp, A. D., and Wodzicki, K. R.: Response of the intertropical convergence zone to climate change: Location, width, and strength, Current Climate Change Reports, 4, 355–370, 2018.
- Li, J., Wang, W.-C., Dong, X., and Mao, J.: Cloud-radiation-precipitation associations over the Asian monsoon region: an observational analysis, Climate Dynamics, 49, 3237–3255, 2017.
- Neelin, J. D. and Held, I. M.: Modeling tropical convergence based on the moist static energy budget, Monthly Weather Review, 115, 3–12, 1987.
- Rose, B. E.: CLIMLAB: a Python toolkit for interactive, process-oriented climate modeling., J. Open Source Softw., 3, 659, 2018.
- Stephens, G. L., Shiro, K. A., Hakuba, M. Z., Takahashi, H., Pilewskie, J. A., Andrews, T., Stubenrauch, C. J., and Wu, L.: Tropical deep convection, cloud feedbacks and climate sensitivity, Surveys in Geophysics, 45, 1903–1931, 2024.
- Wang, F., Zhang, H., Chen, Q., Zhao, M., and You, T.: Analysis of short-term cloud feedback in East Asia using cloud radiative kernels, Advances in Atmospheric Sciences, 37, 1007–1018, 2020.